# Bread By-Product and Maize Silage as Alternative Ingredient Feeds for Production of *Tenebrio molitor* Larvae in High-Concentrate Substrates

**DOI:** 10.3390/ani14233505

**Published:** 2024-12-04

**Authors:** Guillermo Fondevila, Ana Remiro, Sara Remón, Manuel Fondevila

**Affiliations:** Departamento de Producción Animal y Ciencia de los Alimentos, Instituto Agroalimentario de Aragón (IA2), Universidad de Zaragoza-CITA, M Servet 177, 50013 Zaragoza, Spain; guillermofondevila@gmail.com (G.F.); a.remiro.romera@gmail.com (A.R.); remon@unizar.es (S.R.)

**Keywords:** insect feeding, bread by-product, maize silage, larval growth

## Abstract

Raising insects has become an interesting alternative to provide with high quality nutrients for animal feeding, but its potential relies on the reduction of production costs. In this respect, bread by-product and maize silage were proposed to substitute conventional feed ingredients such as wheat grain and wheat bran in isonitrogenous substrates for Tenebrio molitor larvae. The inclusion of bread by-product in the substrate maintained larval growth rate and composition, although at the expense of higher substrate intake, thus reducing efficiency. Maize silage, however, reduced larval growth and protein deposition, a result of a lower substrate intake and poorer feed efficiency. Both bread by-product and maize silage can be potentially used for feeding T. molitor larvae, but the impairments in growth performance in case of maize silage should be considered to optimize production costs.

## 1. Introduction

Insect production for animal feeding has arisen as a reliable and sustainable alternative to traditional protein sources in the present century [1,2,3]. In this regard, a great effort has been made to increase knowledge of the optimal feeding conditions of promising insect species such as *Tenebrio molitor*. It is assumed that *T. molitor* larvae can be intensively raised with mixtures of cereal flours and wheat bran [4,5]. The use of other ingredients, such as agro-industrial by-products and residues from human-food processing have been proposed to reduce feeding costs [6,7,8,9], can contribute to reduce the environmental impact of agriculture [10]. However, the high fiber content in most of these ingredients alters substrate composition and affects larval performance and nutrient content. Consequently, further research is needed to evaluate the suitability of any potential ingredient to be included in the substrate [9,11,12]. Besides, environmental seasonality according to the region of production might restrict the availability of some of these resources, thus limiting their utilization as feeds. Therefore, it is necessary to find alternative ingredients to reduce feeding costs for insect production and ensure the availability of adequate substrates throughout the year.

Bread crumbs are an important waste from the bakery industry that could be advantageously used as an ingredient for animal feeding, contributing to the reduction of the environmental impact of the food chain [13]. Even though nutrient composition of bread by-product is highly variable due to the original material used [14,15], its high starch content (approximately 700–730 g/kg; [16]) and its high digestibility, which may result from the temperature applied during the baking process, make it a suitable energy source for *T. molitor* larvae. Although some publications reported reduced consumption of bread by-product by larvae [17], this reduction in substrate intake has been attributed to the presence of specific substances such as cinnamon when mixed with cookies and pastry by-products [12]. In the absence of these minor compounds, no further impairments on growth performance should be expected from the inclusion of bread by-product in the substrate for *T. molitor* larvae.

Maize silage is a forage widely used in ruminant feeding worldwide, with a high energy value due to the greater starch content compared to other fibrous ingredients [18,19]. For silage production, the whole plant of maize is harvested before grain matures at a 30 to 35% dry matter (DM) content and then it is chopped, pressed, sealed, allowed for anaerobic fermentation, and eventually dried to prevent mold development [20,21]. This process preserves the quality and nutritive value of the fresh product and ensures a long-term availability preventing feed seasonality [20,22]. Consequently, maize silage could be considered as an interesting alternative ingredient for industrial insect feeding. Previous research [23,24] has shown improved biomass production by using fermented feeds for raising *H. illucens* larvae. However, substrates with pH values below 3.5 may significantly decrease feed intake and impair larval performance [25], suggesting that the low pH of maize silage might promote some extent of rejection by *T. molitor* larvae.

We hypothesized that bread by-product can be a good alternative to cereals as energy source, and maize silage can be an economic, whole-year available fibrous ingredient for rearing *T. molitor* larvae. Furthermore, because of their different nutrient composition, both can be complementary in substrate formulation. Therefore, the objective of this study was to assess the potential value of bread by-product and maize silage as alternative feeds for intensive production of *T. molitor* larvae, in substitution of conventional ingredients in isonitrogenous substrates, focusing on their effects on productive performance and nutrient production.

## 2. Materials and Methods

### 2.1. Ingredients and Experimental Substrates

Representative samples of pre-cooked bread and maize silage were provided by NOVAPAN (La Puebla de Alfindén, Zaragoza, Spain) and LIVERCO S.L. (Buñuel, Navarra, Spain), respectively. The bread by-product used was discarded pre-cooked bread from industrial production of breadcrumbs that underwent a second baking step in a convection oven at 180 °C for 20 min, followed by a grating process, and sieved to a 1–3 mm size. Upon arrival, the pre-cooked bread had a DM content of 696 g/kg, while the maize silage had a DM content of 365 g/kg, a pH of 3.84, and an ammonia concentration of 6.6 g/kg DM. Due to the high initial moisture content of these feeds, both ingredients were dried prior to their inclusion in the substrates to prevent mold growth during the experiment, reaching 827 and 920 g DM/kg for bread by-product and maize silage, respectively. Four experimental treatments were defined based on the ingredients used in the substrate: from a control substrate (CTL) consisting of 620 g wheat grain, 340 g wheat bran, and 40 g soybean meal per kilogram, three experimental diets were formulated in which wheat grain and wheat bran were partially substituted by bread by-product (BBP) or bread by-product combined with either a low (170 g/kg, MSL) or a high (310 g/kg, MSH) proportion of maize silage. Nutrient composition (crude protein, CP, neutral detergent fiber, NDF, and starch, on DM basis) of the common ingredients was: wheat bran, 167 g/kg CP, 308 g/kg NDF, and 287 g/kg starch; wheat grain, 107 g/kg CP, 89 g/kg NDF, and 680 g/kg starch; soybean meal, 540 g/kg CP and 126 g/kg NDF. Levels of inclusion of the tested feeds were fixed to allow to diets formulation containing two ranges of NDF and a starch content from 520 to 580 g/kg DM to avoid any energy deficiency, and soybean meal was included at different proportions to adjust the crude protein (CP) content of the substrates to 155 g DM/kg. All ingredients were ground through a 2 mm sieve in a hammermill (RETSCH 5657, Haan, Germany) before being used as insect substrate. The chemical composition of the ingredients and substrates under test is presented in Table 1.

### 2.2. Experimental Procedures

Prior to the start of the experiment, hatched larvae were offered a substrate based on 1:1 wheat grain and wheat bran. Four trays (15 × 9 × 6 cm) were set up for each experimental treatment, each tray including 24 g of substrate feed. Larvae of approximately 6 weeks of age from our colony were sieved through a 1-mm screen and those with 6 to 8 mm and 16.1 ± 0.67 mg weight were considered. Among them, 640 larvae were randomly selected and 40 were included per tray to reach a density of 30 larvae per dm^2^. The experiment lasted for 49 days prior to larval pupation, in which the larvae received their corresponding experimental substrate in an environmentally controlled room, with average minimum and maximum temperature and relative humidity values of 23 ± 1.9 and 28 ± 1.5 °C and 39 ± 8.6 and 48 ± 9.3%, respectively. The larvae were maintained in darkness throughout the experiment, except for sampling procedures.

In addition to the substrate, 1.8 to 2.2 g of fresh carrot (91 g DM/kg) was provided to each tray twice per week as a source of water for larvae. Any piece of carrot remaining in the trays was removed before adding the new portion. Weekly, larvae and substrate remaining per tray were weighed (METTLER Toledo PG5002S, Greifensee, Switzerland) and larval mortality was calculated by counting live larvae, in order to estimate mass gain (total increase in larval weight), individual larval growth (considered as individual larval weight gain, once corrected from mortality), substrate intake (as the difference between initial substrate and final residue), and feed-to-gain ratio (F:G, calculated as the amount of substrate consumed per unit of weight gained). The estimation of feed intake was not corrected for the frass remaining in the residual substrate, since the finer components of the feed residue could not be separated from the frass, as it especially occurs for ground wheat grain, in which 26% of particles were below 0.3 mm [26]. At the end of the experimental period, all larvae remaining on each tray were weighed and collected for chemical analyses. Because of the limited amount of larval sample obtained per tray at the end of the experiment, larvae from all the replicates within each treatment were pooled for further chemical analysis. Then, larval samples were frozen at −80 °C and lyophilized (LyoBeta 25, TELSTAR, initial temperature −45 °C; 48 h at −25 °C, 55 μbar; 24 h at 25 °C) and analyzed in the laboratory to estimate total larval production of DM, CP, ether extract (EE), and acid detergent fiber (ADF) per tray. Feed residues per tray were also sampled for the determination of DM, CP, starch, and NDF contents, and the results were used to calculate individual nutrient consumption. Analyses for ingredients, substrates, larvae, and residues were performed in duplicate.

### 2.3. Chemical Analyses

Ingredients and substrates were analyzed for DM, organic matter (OM), CP (as N × 6.25), and EE following the AOAC [27] procedures (methods ref. 934.01, 942.05, ref. 976.05, and ref. 2003.05, respectively). The concentration of NDF was analyzed as described by Mertens [28] in an Ankom 200 Fiber Analyzer (Ankom Technology, NY, USA), using α–amylase and sodium sulfite, and results were expressed exclusive of residual ashes. Total starch content was determined enzymatically from samples ground to 0.5 mm using a commercial kit (Total Starch Assay Kit K-TSTA 07/11; Megazyme, Bray, Ireland). The DM content of larvae was determined after lyophilization. Larvae were also analyzed for OM, CP, and EE, as indicated previously, and for ADF (ref. 973.18, [27]).

### 2.4. Statistical Analyses

The effects of treatment on productive performance, nutrient intake, and total production of DM, CP, EE, and ADF were analyzed by ANOVA (SAS Institute, 2019) according to the following model:$$yij=\mu +\alpha i+\beta (xij-\bar{x})+\varepsilon ij$$where *μ* is the mean, *αi* is the treatment effect as fixed effect (n = 4), *β* is the regression coefficient that relates *yij* with *xij* as the covariate, $\bar{x}$ is the mean for the covariate, and *εij* is the random error. The tray was considered as the experimental unit (n = 4). Initial larval weight was considered as a covariable for correcting its effect on experimental results. All parameters were explored for normality using the Shapiro–Wilk test and results were contrasted by the Kruskall–Wallis nonparametric test. Weekly pattern of larval weight according to treatment was tested as indicated by Littell et al. [29], considering the sampling date as repeated measures. When significant, treatment differences were compared by the Tukey test. Differences with *p* < 0.05 and between *p* = 0.10 and *p* ≥ 0.05 were considered as significant or as a trend for significance, respectively.

## 3. Results

Initial larval weight per tray did not differ among treatments (0.643, 0.557, 0.683, and 0.663 g for CTL, BBP, MSL, and MSH, respectively; SEM = 0.052; *p* = 0.363). The weekly pattern of individual larval weight along the experimental period is shown in Figure 1. A significant interaction between treatment and sampling date was detected (*p* < 0.001). No differences among treatments were observed for larval weight from 0 to 21 days (*p* > 0.10). From 28 to 42 days, however, larvae fed CTL showed greater individual weight than those fed MSL and MSH (*p* < 0.05), with larvae fed BBP being intermediate. At the end of the experiment (49 days), CTL and BBP substrates resulted in heavier larvae compared to MSL and MSH (*p* < 0.01).

The effects of the experimental substrate fed to the larvae on productive performance from 0 to 49 days is shown in Table 2. Overall mortality was 14% and was numerically lowest for BBP, but no statistical differences were observed (*p* = 0.261). Total larval mass gained per tray was higher (*p* < 0.001) for CTL and BBP than for treatments including maize silage in substitution of wheat grain and wheat bran (MSL and MSH). As a result, individual larval growth corrected by the number of living larvae was greater for CTL and BBP than for MSL and MSH (*p* < 0.001). Substrate intake in fresh basis was greater for BBP than for CTL, MSL, or MSH (*p* < 0.001). As a result, F:G was lowest for CTL followed by BBP, and lower for BBP than for MSL and MSH (*p* < 0.001).

Total DM intake per tray (Table 3) was greater for BBP and CTL than for MSH and MSL (*p* < 0.001). Starch consumption per tray was greater for larvae fed BBP than for those given MSH, with larvae fed CTL and MSL being intermediate (*p* < 0.001). Total consumption of CP was greatest for BBP, intermediate for CTL and MSL, and lowest for MSH (*p* < 0.05), whereas NDF intake was greatest for CTL, intermediate for BBP and MSH, and lowest for MSL (*p* < 0.001). The analyzed chemical composition of pooled samples of larvae fed CTL, BBP, MSL, and MSH after 49 days was, respectively: DM, 379, 392, 382, and 400 mg/g; CP, 486, 445, 470, and 474 mg/g DM; EE, 353, 410, 377, and 370 mg/g DM; and ADF, 57, 60, 65, and 69 mg/g DM, respectively. These values were used to calculate total larval production per tray (also in Table 3). Total production of DM, CP, EE, and ADF per tray was higher (*p* < 0.001) in larvae fed BBP and CTL than in those fed both silage treatments in all cases.

## 4. Discussion

Mortality of larvae throughout the experiment did not differ among treatments, with values ranging from 5 to 17%, which are within the range of values observed in previous research [7,12,30]. It has been stated that a low dietary protein level can be a mortality factor for *T. molitor* larvae [7,12]; moreover, an increased larval mortality due to feeding cookie remains was attributed to the presence of cinnamon and clove [12]. Any potential effects of these can be discarded in the present experiment, as substrate CP concentration was over the larval requirements [30] and bread by-product was free of spices. The lack of differences in mortality among treatments in this work can be partly attributed to the high variability recorded for this parameter (coefficient of variation of 0.51). Results from the present experiment indicate that *T. molitor* maintains a relatively constant growth pattern along the last phase of larval growth until the pupal stage (from 28 to 49 days of the study), which agrees with that observed in previous experiments from our group [26] and others [17,25].

The information available on the effects of the inclusion of bread by-product in the diet of *T. molitor* larvae is scarce. Bread by-product is based on ground remains of wheat bread that have not been commercialized for human consumption, which in turn has been subjected to heat treatment along the baking process that is considered to increase starch digestibility. Previous research included bread by-products in the substrate in combination with different ingredients with variable nutrient composition, thus not allowing for sound conclusions about its potential use. For example, Montalbán et al. [31] included bread remains in diets for *T. molitor* larvae, but its level of inclusion (from 20 to 85%) was inversely related to the substrate protein content (from 185 to 263 g/kg CP), and thus, it is difficult to associate the response to a single cause. Similarly, Mancini et al. [17] observed poorer larval development in substrates based on bread, cookie remains, or a 1:1 mixture of both ingredients, compared to brewery-spent grains as a source of protein. However, substrate CP content recalculated as N × 6.25 varied widely (from 65 to 193 g/kg), and consequently, performance results cannot be directly associated to the ingredients itself. Oonincx et al. [7] tested two diets for *T. molitor* larvae, including 50% bread together with other ingredients such as cookie remains, potato peelings, and beet molasses, and observed a considerably higher F:G ratio with respect to control diets. However, ingredient and nutrient composition in this research were largely variable and the results cannot be directly associated to the effect of a single ingredient. Also, van Broekhoven et al. [12] tested a mixture of several bakery by-products (bread, cookies, etc.), but results were not comparable to those presented in the current research because of differences in the characteristics of the tested products. In this regard, these authors suggested that certain antinutritional effects associated to cinnamon and clove, not present in the bread by-product used in the present study, could have affected their results. In the current research, in which the nutrient composition of the substrates was balanced in terms of CP and starch contents, the inclusion of 690 g/kg of bread by-product in substitution of wheat grain and part of the wheat bran and soybean meal from the CTL diet, did not alter growth performance and nutrient deposition with a numerical increase in larval survival. These results suggest that bread by-product is an economic and sustainable resource that could be satisfactorily included in substrates for *T. molitor* larvae to optimize larval growth.

Larval production was higher for CTL and BBP than for the maize silage containing diets, probably a result of the reduction observed in substrate intake with MSH and MSL. Differences can hardly be explained by variations in the nutrient content among diets, as all substrates were formulated to contain a similar CP content and starch concentration differed in less than 60 g/kg between extreme values. Similarly, variation in EE or NDF contents among diets do not appear to match the differences observed in nutrient intake. Consequently, the reduction in substrate consumption in MSL and MSH vs. CTL and BBP must be considered with caution, and may be associated to a certain level of rejection of maize silage when included in the substrate instead of wheat grain and wheat bran. Total intakes of starch, CP, and NDF were greater for larvae fed BBP and CTL than for those fed substrates that included maize silage despite its level of inclusion. Furthermore, the higher starch and lower NDF content contents in BBP and MSL with respect to CTL and MSH substrates did not necessarily match with a greater substrate intake or lower F:G ratio, suggesting that other factors than carbohydrate composition may affect growth performance and substrate acceptance by larvae. In this respect, Fondevila et al. [26] did not observe differences in substrate intake or growth performance among diets with starch to NDF ratios ranging from 1.0 to 1.4. In the present experiment, however, the starch proportion was much higher (starch to NDF ratios from 3.1 to 4.9), which in turn justifies the higher larval growth recorded in the present experiment compared to that of Fondevila et al. [26]. In this context, the lower performance recorded with maize silage cannot be directly associated to differences in the energy, CP, or fiber contents in these diets, and consequently, it might be associated to an intrinsic ingredient effect.

In ruminant livestock, the microbial fermentation occurring along the ensiling process increases fiber digestibility [32,33], making nutrients, both polysaccharides and proteins, more accessible. There are several examples of a positive effect of using fermented feeds for raising insect larvae. Howdeshell and Tanaka [23] and Wong et al. [34] reported that the microbial fermentation occurring in dry distillers’ grains with solubles (DDGS) and coconut endosperm silage production, respectively, increased substrate intake, growth performance, and nutrient utilization by black soldier fly (*Hermetia illucens*) larvae. Similarly, Kuttiyatveetil et al. [24] observed that growth of *H. illucens* larvae increased by 20 to 30% with fermented by-products (borage and flaxseeds) with respect to unfermented substrates, with an increase in larval protein deposition. In *T. molitor* larvae reared on brewers’ spent grains, substrate intake was reduced by 17% compared to a wheat bran diet, but larval growth was not affected, and in fact, larvae raised with brewers’ spent grains retained more protein and less fat [35]. Conversely, Zhang et al. [36] reported that larvae growth when fed spirit distillers’ grains was 53% of that achieved with wheat bran. It could be discussed that the low pH of the maize silage (3.84) might alter the feeding pattern and reduce substrate intake and larval performance. In this regard, Ma et al. [37] reported that feeding behavior of *H. illucens* larvae was influenced by the pH of the substrate, and reported lower growth performance at pH values of 2.0 and 4.0 with respect to 6.0 and above. In *T. molitor* larvae, Coudron et al. [25] speculated that an acidic substrate might drop the pH along the digestive tract, and thus, reduce activity of digestive enzymes. However, these authors did not observe any negative effect once the pH was maintained over 3.5. Although the pH of the whole mixed substrate was not measured in the present study, it can be assumed that it should be buffered by the other ingredients in the mixture. Nevertheless, the impairment in substrate intake as a result of the low pH of the silage cannot be disregarded.

Previous research suggests that maize-derived products might impair larval performance when included in the substrate. In this respect, Stull et al. [38] observed that growth performance of *T. molitor* larvae dropped from 6.65 to 2.38 and 1.60 mg/d when a 50:45:5 mixture of wheat bran, oats, and brewer yeasts was substituted by others including 40 or 100% maize stovers. Similarly, Pascual et al. [39] recorded lower larval growth when fed maize grain or maize by-products (gluten feed, hominy feed) compared with barley grain and wheat bran. In fact, these authors reported that nutrient digestibility for gluten feed, a maize by-product with a similar fiber content than maize silage, was lower compared to other raw materials. Besides, results from our group [40] indicated that feed intake and growth performance of *T. molitor* larvae were consistently lower when maize grain was included in the substrate compared to wheat or barley, either when fed as the unique source of nutrients or as the main source of energy in isonitrogenous diets. To some extent, this information suggests that the physical characteristics of maize, including the starch structure of the grain, more vitreous and with larger granules [41], as well as the fibrous components of the plant, such as maize bran, leaves, and stalks, might promote certain rejection for consumption in *T. molitor* larvae. Factors related to the particular anatomy of the mandibles and mouthparts of the larvae or simply a preference behavior for other ingredients might explain, at least in part, the impairments in feeding performance with maize products observed in vivo [42]. To our knowledge, there are no reports in the literature on the inclusion of maize silage in insect diets, but the results obtained in the current research suggest that the inclusion of this forage in the substrate might reduce growth performance and impair F:G ratio of *T. molitor* larvae. Consequently, using MSL and MSH in production of *T. molitor* larvae could be economically feasible provided that costs of these diets were 60–65% of that of CTL-type substrates; however, the reduction of larval growth and substrate efficiency when maize silage was used has to be considered.

Results on the chemical composition of *T. molitor* larvae at their latest instar stage indicate that minor differences might be expected in CP in EE content within the dietary conditions considered in the current research. Interestingly, although differences were apparently low, CP proportion was numerically lower in larvae fed BBP, with a subsequent higher fat content compared to those fed other treatments. In this respect, Mancini et al. [17] only recorded differences in protein or fat composition among diets with extreme CP levels (6.6 vs. 17.9%). Also, Plonquet et al. [40] observed that larval composition varied when fed cereals (wheat, barley, and maize) as the unique source of nutrients in the substrate, but these differences were reduced when combined with other ingredients in isonutritive diets. The wide differences observed in the literature on the chemical composition of *T. molitor* larvae are probably related to variations in the ingredient and chemical composition of the substrates, but also to differences in the methodologies applied (environmental conditions, age at harvesting, analytical procedures, etc.). In this context, further research is needed to provide analytical data to better understand the influence of the ingredients included in the diet on the nutrient content of *T. molitor* larvae.

## 5. Conclusions

High-concentrate diets based on wheat or bread by-product promoted similar growth performance of *T. molitor* larvae when combined with wheat bran, even at high inclusion levels. In contrast, the inclusion of maize silage in the substrate at 170 to 310 g/kg in substitution of wheat grain and wheat bran reduced larval growth and impaired F:G, as a result of a significant reduction in feed intake, and hence, its proportion in diet should be moderate. Bread by-product and maize silage can be valid feed alternatives to raise *T. molitor* larvae because of their low price and potential market availability. However, a single experiment and the scarce literature available do not allow for extracting sound practical recommendations on the use of these feeds in productive conditions. Furthermore, variations in larval performance when maize silage is included in the substrate should be considered to optimize production costs.

## Figures and Tables

**Figure 1 animals-14-03505-f001:**
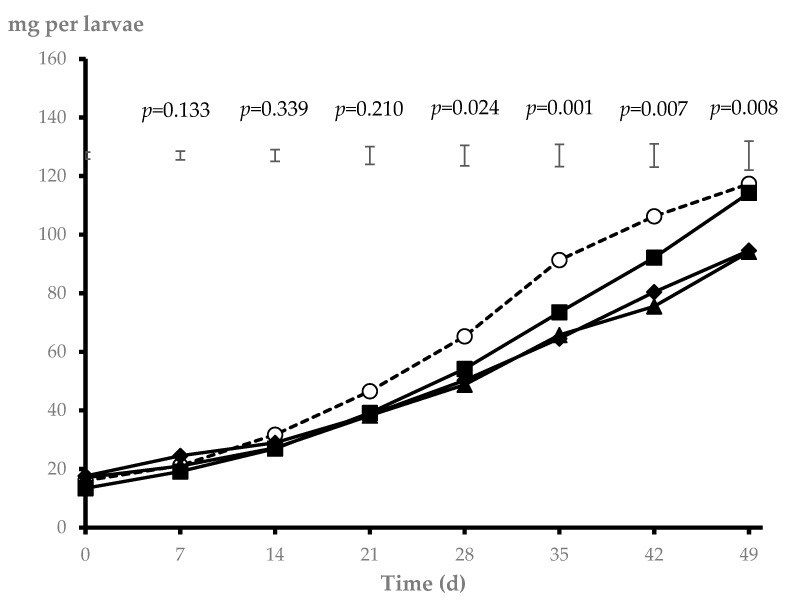
Larval weight pattern (mg) from 0 to 49 days for *T. molitor* larvae fed the control substrate (CTL, ○), substrates including bread by-product (BBP, ■), or bread by-product with low (MSL, ▲) or high (MSH, ◆) proportions of maize silage (interaction treatment × sampling date, *p* < 0.001). The *p*-values and standard error of the means (vertical bars) for the statistical comparison within sampling date are also shown (n = 4).

**Table 1 animals-14-03505-t001:** Ingredient composition of experimental substrates (g/kg fresh matter) and chemical composition (g/kg dry matter, DM) of the tested ingredients and mixed feeds.

	Bread By-Product	Maize Silage	CTL	BBP	MSL	MSH
Ingredients:						
Wheat grain			620	---	---	---
Wheat bran			340	300	---	---
Soybean meal			40	10	90	90
Bread by-product			---	690	740	600
Maize silage			---	---	170	310
Chemical composition:						
Dry matter	827	920	891	848	847	860
Organic matter	974	958	972	967	966	964
Crude protein	142	67	153	154	165	154
Ether extract	4	28	26	16	9	13
Starch	720	324	520	577	578	524
Neutral detergent fiber	47	396	165	130	119	170

CTL, control substrate; BBP, substrate including bread by-product; MSL, substrate including bread by-product and a low proportion of maize silage; MSH, substrate including bread by-product and a high proportion of maize silage.

**Table 2 animals-14-03505-t002:** Mortality (%), mass gained (g per tray), individual larval growth (mg/d per larvae), feed intake (g per tray), and feed-to-gain ratio (F:G, g/g) of *T. molitor* larvae fed the control substrate (CTL), substrates including bread by-product (BBP), or bread by-product with low (MSL) or high (MSH) proportions of maize silage for 49 days.

	CTL	BBP	MSL	MSH	SEM	*p*-Value
Mortality	15.0	5.4	16.8	17.1	3.57	0.261
Mass gained	4.00 a	3.95 a	2.59 b	2.51 b	0.169	<0.001
Individual growth	2.49 a	2.18 a	1.66 b	1.65 b	0.066	<0.001
Feed intake	5.48 b	7.41 a	5.76 b	5.47 b	0.197	<0.001
F:G	1.373 c	1.893 b	2.226 a	2.160 a	0.0573	<0.001

Within rows, different letters indicate significant differences (*p* < 0.05). SEM: standard error of means (n = 4); DM: dry matter.

**Table 3 animals-14-03505-t003:** Estimated nutrient intake and nutrient production of *T. molitor* larvae after 49 days receiving the control substrate (CTL), substrates including bread by-product (BBP), or bread by-product with low (MSL) or high (MSH) proportions of maize silage.

	CTL	BBP	MSL	MSH	SEM	*p*-Value
Total intake (mg per tray)						
DM	4273 a	4997 a	3286 b	3337 b	182.5	<0.001
Starch	2222 b	2869 a	1945 bc	1747 c	100.1	<0.001
CP	656 ab	770 a	562 bc	514 c	29.1	0.021
NDF	707 a	661 ab	374 c	568 b	26.8	<0.001
Production (mg per tray)						
DM	1516 a	1546 a	989 b	1007 b	64.5	<0.001
CP	737 a	691 a	465 b	477 b	30.9	<0.001
EE	535 a	631 a	374 b	373 b	23.5	<0.001
ADF	87.6 a	93.5 a	64.5 b	69.4 b	3.92	0.003

Within rows, different letters indicate significant differences (*p* < 0.05). SEM: standard error of means (n = 4); DM: dry matter; CP: crude protein; NDF: neutral detergent fiber; EE: ether extract; ADF: acid detergent fiber.

## Data Availability

Data are contained within the article.

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
