# Peer review of "Bread By-Product and Maize Silage as Alternative Ingredient Feeds for Production of *Tenebrio molitor* Larvae in High-Concentrate Substrates"

_animals, 2024, doi:10.3390/ani14233505_

Round 1
Reviewer 1 Report
Comments and Suggestions for Authors
Congratulations on the execution of the study. The work is well written and structured, but it needs some improvements:
In the introduction, please add new studies; the literature used is very scarce. I suggest reading the review: https://doi.org/10.3390/insects15080619, it can be useful.
I also suggest others:
Gasco, L.; Biasato, I.; Dabbou, S.; Schiavone, A.; Gai, F. Animals fed insect-based diets: State-of-the-art on digestibility, performance and product quality. Animals 2019, 9, 170. [Google Scholar] [CrossRef] [PubMed]
DiGiacomo, K.; Leury, B.J. Insect meal: A future source of protein feed for pigs? Animal 2019, 13, 3022–3030. [Google Scholar] [CrossRef] [PubMed] Makkar, H.P.; Tran, G.; Heuzé, V.; Ankers, P. State-of-the-art on use of insects as animal feed. Anim. Feed Sci. Technol. 2014, 197, 1–33. [Google Scholar] [CrossRef] Sorjonen, J.M.; Valtonen, A.; Hirvisalo, E.; Karhapää, M.; Lehtovaara, V.J.; Lindgren, J.; Marnila, P.; Mooney, P.; Mäki, M.; Siljander-Rasi, H.; et al. The plant-based by-product diets for the mass-rearing of Acheta domesticus and Gryllus bimaculatus. PLoS ONE 2019, 14, e0218830. [Google Scholar] [CrossRef] [PubMed] Gasco, L.; Dabbou, S.; Trocino, A.; Xiccato, G.; Capucchio, M.T.; Biasato, I.; Dezzutto, D.; Birolo, M.; Meneguz, M.; Schiavone, A.; et al. Effect of dietary supplementation with insect fats on growth performance, digestive efficiency and health of rabbits. J. Anim. Sci. Biotechnol. 2019, 10, 4. [Google Scholar] [CrossRef] [PubMed] Van Huis, A. Prospects of insects as food and feed. Agric Org. 2021, 11, 301–308. [Google Scholar] [CrossRef] Lähteenmäki-Uutela, A.; Grmelová, N.; Hénault-Ethier, L.; Deschamps, M.H.; Vandenberg, G. W.; Zhao, A.; Zhang, Y.; Yang, B.; Nemane, V. Insects as food and feed: Laws of the European union, United States, Canada, Mexico, Australia, and China. Eur. Food Feed Law Rev. 2017, 12, 22–36. [Google Scholar] DiGiacomo, K.; Akit, H.; Leury, B.J. Insects: A novel animal-feed protein source for the Australian market. Anim. Prod. Sci. 2019, 59, 2037–2045. [Google Scholar] [CrossRef] Cappellozza, S.; Leonardi, M. G.; Savoldelli, S.; Carminati, D.; Rizzolo, A.; Cortellino, G.; Terova, G.; Moretto, E.; Badaile, A.; Concheri, G.; et al. A first attempt to produce proteins from insects by means of a circular economy. Animals 2019, 9, 278. [Google Scholar] [CrossRef] [PubMed] Apri, A.D.; Komalasari, K. Feed and animal nutrition: Insect as animal feed. In IOP Conference Series: Earth and Environmental Science; IOP Publishing: Bristol, UK, 2020; Volume 465, p. 012002. [Google Scholar] Jayanegara, A.; Novandri, B.; Yantina, N.; Ridla, M. Use of BSF larvae (Hermetia illucens) to substitute soybean meal in ruminant diet: An in vitro rumen fermentation study. Vet. World 2017, 10, 1439. [Google Scholar] [CrossRef] [PubMed] Jayanegara, A.; Yantina, N.; Novandri, B.; Laconi, E.B.; Ridla, M. Evaluation of some insects as potential feed ingredients for ruminants: Chemical composition, in vitro rumen fermentation and methane emissions. J. Indones. Trop. Anim. Agric. 2017, 42, 247–254. [Google Scholar] [CrossRef] Rashmi, K. M.; Chandrasekharaia, M.; Soren, N.M.; Prasad, K.S.; David, C.G.; Thirupathaiah, Y.; Shivaprasad, V. Effect of dietary incorporation of silkworm pupae meal on in vitro rumen fermentation and digestibility. Indian J. Anim. Sci. 2018, 88, 731–735. [Google Scholar] [CrossRef]
Toral, P.G.; Hervás, G.; González-Rosales, M.G.; Mendoza, A.G.; Robles-Jiménez, L.E.; Frutos, P. Insects as alternative feed for ruminants: Comparison of protein evaluation methods. J. Anim. Sci. Biotechnol. 2022, 13, 21. [Google Scholar] [CrossRef] [PubMed]
- Highlight the main precautions that should be taken when feeding the larvae. Are there any restrictions?
- Add the basal composition of the wheat bran and other ingredients in the introduction to guide the reader.
- What is the main hypothesis of the study? Make it clear.
- Avoid too many explanations in the objective; make it more aligned with the title.
Material and methods
- What is the pre-cooking temperature?
- What are the inclusion and exclusion criteria for choosing the larvae?
- Were there 40 larvae per treatment? Explain better.
- Add photos of the weighing and measurements of them.
- Table 1 should appear at the time it is mentioned in the text.
- Explain better how the weight gain of the larvae was measured. If possible, add images in panel format to help us better understand your study.
- Please add all the brands and models of equipment you mention so that other studies can replicate them.
- Please divide topic 2.2 into two, as it is tiring and exhausting as it is.
- Divide topic 2.3 into two, leaving only chemical analyses and another for statistics, these are important topics; - About the statistics, I miss a statistical model to make it easier to visualize, since you have 4 treatments...
Results
- Whenever you talk about a difference or not in p-value, please cite it.
- Do not use P or p when talking about p-value.
Discussion
- Please describe more about the factors that can lead to the death of these larvae.
- Discuss the adverse effects that bread food can cause on the larvae.
- Present the main limitations of the study, for example, was the number of larvae sufficient?
Conclusion
The conclusion is assertive, but add a paragraph of recommendations.
Author Response
Congratulations on the execution of the study. The work is well written and structured, but it needs some improvements:
In the introduction, please add new studies; the literature used is very scarce. I suggest reading the review: https://doi.org/10.3390/insects15080619, it can be useful.
RESPONSE: Thanks very much for your proposed literature, which includes some papers that we did not have. Two new references have been added to support the first general statement on the use of insect products as feeds. However, we want to avoid being exhaustive in citing references that are not directly related to the topic of the study, which is the use of preserved forages and by-products (or specifically, maize silage and bread by-product) for rearing Tenebrio molitor larvae.
- Highlight the main precautions that should be taken when feeding the larvae. Are there any restrictions?
RESPONSE: I agree it should be interesting to add information about the care that should be taken in feeding T. molitor larvae in terms of nutrient composition and presentation form of ingredients; however, this would extend the length of the Introduction section, that already has 45 lines. This information is already mentioned in part regarding the potential effects of the tested ingredients.
- Add the basal composition of the wheat bran and other ingredients in the introduction to guide the reader.
RESPONSE: For text homogeneity, the required information (analysed nutrient composition of the wheat bran, wheat grain and soybean meal used) is now summarised in Material and methods (Subsection 2.1).
- What is the main hypothesis of the study? Make it clear.
RESPONSE: The specific hypothesis has been now included in the last paragraph of the Introduction section.
- Avoid too many explanations in the objective; make it more aligned with the title.
RESPONSE: Writing of the objectives have been simplified.
Material and methods
- What is the pre-cooking temperature?
RESPONSE: This information has been added in Subsection 2.1.
- What are the inclusion and exclusion criteria for choosing the larvae?
RESPONSE: As stated in text, larvae were firstly chosen by age, and then were randomly taken for a homogeneous size, as they were sieved through a 1 mm screen
- Were there 40 larvae per treatment? Explain better.
RESPONSE: The selection and distribution of larva is now clarified in text (Subsection 2.2). Forty larvae were included per tray, and four trays were assayed per treatment, so in total 160 larvae were assayed per treatment.
- Add photos of the weighing and measurements of them.
RESPONSE: Larvae were weighed under conventional procedures: total larvae from a tray were picked up manually, counted and weighed in group in a Mettler scientific balance. I do not see how a photo of the weighing procedures should improve the information.
- Table 1 should appear at the time it is mentioned in the text.
RESPONSE: Table 1 was placed in the next page after citing (at the end of the paragraph) because it does not entirely fit in the page where it is cited.
- Explain better how the weight gain of the larvae was measured. If possible, add images in panel format to help us better understand your study.
RESPONSE: Larvae were weighed under conventional procedures: total larvae from a tray were picked up manually, counted and weighed in group in a Mettler scientific balance. We do not have photos of the weighing procedure, but anyway I do not see how a photo of it should improve the information.
- Please add all the brands and models of equipment you mention so that other studies can replicate them.
RESPONSE: Information specifying the used equipment has been added in the new version.
- Please divide topic 2.2 into two, as it is tiring and exhausting as it is. Divide topic 2.3 into two, leaving only chemical analyses and another for statistics, these are important topics; - About the statistics, I miss a statistical model to make it easier to visualize, since you have 4 treatments...
RESPONSE: Subsection 2.2 is only 20 lines long, and its structure is homogeneous as it reports procedures of a single experiment, and we cannot find the place to divide it into two sections. However, it has been divided in two paragraphs for an easier reading. Subsection 2.3 has been divided into chemical (2.3) and Statistical (2.4) analysis. The model for statistical analysis has been added in text.
Results
- Whenever you talk about a difference or not in p-value, please cite it.
RESPONSE: The P-value is given every time there are treatment differences. A P-value has now been added also to non-significant differences in this section, as requested.
- Do not use P or p when talking about p-value.
RESPONSE: Sorry, I cannot understand this comment. P is the standard abbreviation for probability value, and as it is mentioned in literature
Discussion
- Please describe more about the factors that can lead to the death of these larvae.
RESPONSE: A mention to potential causes of mortality has been included at the start of Discussion.
- Discuss the adverse effects that bread food can cause on the larvae.
RESPONSE: The major effect of bread by-products on larvae can be attributed to a reduction in CP content of substrate and the presence of cinnamon and clove in some by-products including cookies, and these have now been mentioned regarding mortality
- Present the main limitations of the study, for example, was the number of larvae sufficient?
RESPONSE: The experimental conditions can be considered standard in terms of environmental parameters, water availability (as carrot provided) and larval concentration, so any limitation can be discarded. Besides, in case of an effect it should be general and should not affect treatment comparison. The study was a prospective assay to evaluate the tested feeds as well as their combination, and for doing so we consider four repetitions per treatment is acceptable. Obviously, further results should be necessary to extract firm conclusions.
Conclusion
The conclusion is assertive, but add a paragraph of recommendations.
RESPONSE: In agreement to the comment above, it is difficult to make practical recommendations on the levels of inclusion of the tested feeds from the observed results only. Further, these should depend on whether the main goal is to achieve a high productivity or to reduce productive costs, which is what is stated in text. However, mentions to the level of inclusion of bread by-product and maize silage in the substrate have been added.
Reviewer 2 Report
Comments and Suggestions for Authors
In general, the proposal is interesting, since the mealworm has a great potential for use in different areas and its massive production with higher yields are interesting to study.
why corn silage is not suitable for insect diets
why insects ingest less diet when corn silage is present, detailing at metabolic level
explain why bread, when present in the diets, presents the same yields as wheat bran or wheat grain.
I suggest to expand in the conclusions some perspectives on how your results can support the development of these larvae and what benefits are provided with your study, as well as what further research is needed.
Author Response
In general, the proposal is interesting, since the mealworm has a great potential for use in different areas and its massive production with higher yields are interesting to study.
why corn silage is not suitable for insect diets
RESPONSE: What is hypothesized in the Introduction section is that its low pH might limit larval intake, as it was previously stated (L73-75 of the former version). Besides, we have previously observed a negative acceptance of maize grain, even if substrate is supplemented with protein (Plonquet et al., 2024). However, we have no clear explanation to justify such effect.
why insects ingest less diet when corn silage is present, detailing at metabolic level
RESPONSE: As it is developed along the Discussion section, we consider two aspects that may affect maize silage intake by larvae: whereas a depressing effect of low pH could be discarded from results from Coudron et al. (2022), a potential negative acceptance of maize products that we have previously observed (Plonquet et al., 2024) cannot be discarded. However, we have no clear explanation to justify such effect.
explain why bread, when present in the diets, presents the same yields as wheat bran or wheat grain.
RESPONSE: Simply because this bread by-product has a high proportion of starch, which renders a high energy input. This argument is cited in Discussion.
I suggest to expand in the conclusions some perspectives on how your results can support the development of these larvae and what benefits are provided with your study, as well as what further research is needed.
RESPONSE: Mentions to the level of inclusion of bread by-product and maize silage in the substrate have been added in conclusions. A sentence has been included to indicate the difficulty in making practical recommendations on levels of inclusion of the tested feeds with the observed results only. Further, these should depend on whether the main goal is to achieve a high productivity or to reduce productive costs, which is what is stated in text.
Reviewer 3 Report
Comments and Suggestions for Authors
The term “corn silage” should be used instead of “maize silage” to align with standard terminology.
Bread by-products and corn silage come from very different industrial sectors (bakery and agriculture), with specific and substantially different production processes and objectives. These differences can make a direct and coherent comparison challenging.
If the goal is to explore the viability of by-products in the feeding of Tenebrio molitor, I suggest that each by-product be addressed in separate studies, allowing for a more detailed and conclusive analysis of each one. Alternatively, if it is necessary to keep both ingredients in the same article, I recommend providing a more robust justification for this choice, perhaps expanding the introduction to discuss the reasons why such distinct by-products were included together and how their differences might impact larval performance in complementary ways.
Lines 82–88: The section mentions the use of “pre-cooked bread” without detailing whether it refers to bread crumbs, unsold bread leftovers, or another type of bakery by-product. I recommend specifying if bread crumbs, non-commercialized leftovers, or another sub-product type were used for greater clarity. Additionally, a brief explanation of the drying process (e.g., temperature, duration) would be helpful, as these factors may affect the nutritional composition and quality of the substrate.
Lines 89–94: The specific proportions for replacing wheat and wheat bran with bread by-product and corn silage (170 g/kg and 310 g/kg) are not justified. Adding a brief explanation of the rationale behind these proportions would help contextualize the experimental design and choice of inclusion levels.
Lines 101–127: This paragraph covers several steps (larvae selection, tray setup, feeding, and monitoring) without a clear sequence. I recommend dividing the paragraph, first describing the selection and setup of larvae and trays, then detailing the provision of carrots and weekly monitoring. This would improve the reading flow and comprehension.
Lines 122–129: Better explain the larval sampling procedure, specifying the exact number of larvae per sample and the reasoning for grouping all larvae from each treatment. This is important to ensure the representativeness of the chemical data.
Line 92: The word "susbstituted" contains a typo. Correct to "substituted."
Lines 146–155: The statistical methods should be presented in a separate section for clarity and better organization.
Discussion
Lines 205–315: The discussion is lengthy and covers multiple points without a clear structure. I suggest organizing the discussion into subtopics to improve clarity. Examples of subtopics could include: "Mortality and Growth," "Effects of Bread By-Product," "Impact of corn Silage and pH," and "Nutritional Composition of Larvae."
Lines 237–243: The discussion mentions that the bread by-product did not alter growth performance and numerically increased survival, but it does not sufficiently emphasize its practical potential. I suggest adding a sentence about how the bread by-product can be an economical and efficient alternative to wheat grain, aligning with the sustainability objectives of the research.
Lines 245–265: Although the negative impact of Corn silage on performance is mentioned, I recommend adding a clear statement that the inclusion of Corn silage reduced substrate efficiency for larval growth. This would help avoid ambiguous interpretations regarding the viability of Corn silage as an alternative ingredient.
Line 248: Change "do not apparently match with differences observed in nutrient intake" to "do not appear to match the differences observed in nutrient intake."
Line 256: "necesssarily" contains a typo. Correct it to "necessarily."
Conclusions
The conclusion mentions that diets with wheat or bread by-product promoted similar growth, but it could emphasize that the bread by-product is an economically viable alternative, especially in comparison to wheat grain. I recommend adding a sentence highlighting the potential of the bread by-product as a sustainable and lower-cost option for feeding T. molitor.
Lines 335–338: Although Corn silage is mentioned as a viable alternative, the results showed that it had a negative impact on growth and feed efficiency. I suggest rephrasing the sentence to make it clear that Corn silage has limitations and that its use may only be viable if adjustments are made in formulation or if it is used at low levels.
Author Response
The term “corn silage” should be used instead of “maize silage” to align with standard terminology.
RESPONSE: Both terms “corn silage” and “maize silage” are used to refer the same feed. As in Europe the term “maize silage” is the most commonly accepted, we prefer to maintain this throughout text.
Bread by-products and corn silage come from very different industrial sectors (bakery and agriculture), with specific and substantially different production processes and objectives. These differences can make a direct and coherent comparison challenging.
RESPONSE: We agree with the reviewer. Both products can be economic alternatives for T. molitor feeding, since they are cheap and potentially available in many regions; besides, their nutritional characteristics are quite different. This is why we choose them to test their possible combination in insect feeding.
If the goal is to explore the viability of by-products in the feeding of Tenebrio molitor, I suggest that each by-product be addressed in separate studies, allowing for a more detailed and conclusive analysis of each one. Alternatively, if it is necessary to keep both ingredients in the same article, I recommend providing a more robust justification for this choice, perhaps expanding the introduction to discuss the reasons why such distinct by-products were included together and how their differences might impact larval performance in complementary ways.
RESPONSE: We partly agree with the Reviewer´s comment. In fact, in our study, bread by-product was individually assayed against conventional ingredients (wheat grain and wheat bran, in treatment BBP vs. CTL). At the same time, we considered both tested feeds could be complementary for larval feeding, since maize silage is a widely available fibrous source that can be an alternative to wheat bran, whereas bread by-product gives energy at a similar extent than cereal grains. A sentence highlighting their complementarity has been added to Introduction.
Lines 82–88: The section mentions the use of “pre-cooked bread” without detailing whether it refers to bread crumbs, unsold bread leftovers, or another type of bakery by-product. I recommend specifying if bread crumbs, non-commercialized leftovers, or another sub-product type were used for greater clarity. Additionally, a brief explanation of the drying process (e.g., temperature, duration) would be helpful, as these factors may affect the nutritional composition and quality of the substrate.
RESPONSE: This product is the first stage of a two-steps bread production, that is commercialised as it for being later cooked by the consumer. Description of its characteristics has been extended in the actual version. In order to avoid confusions, once defined the product is subsequently named “bread by-product”.
Lines 89–94: The specific proportions for replacing wheat and wheat bran with bread by-product and corn silage (170 g/kg and 310 g/kg) are not justified. Adding a brief explanation of the rationale behind these proportions would help contextualize the experimental design and choice of inclusion levels.
RESPONSE: Level of inclusion of tested by-products was adjusted in order to maintain a similar nutrient composition of substrate in terms of crude protein, starch and fibre, as it is now further clarified in text (Subsection 2.1). Because of the high NDF content of maize silage and the lack of information about its use in insect diets, two levels of inclusion were assayed.
Lines 101–127: This paragraph covers several steps (larvae selection, tray setup, feeding, and monitoring) without a clear sequence. I recommend dividing the paragraph, first describing the selection and setup of larvae and trays, then detailing the provision of carrots and weekly monitoring. This would improve the reading flow and comprehension.
RESPONSE: The Subsection 2.2 has been divided into two paragraphs, as recommended
Lines 122–129: Better explain the larval sampling procedure, specifying the exact number of larvae per sample and the reasoning for grouping all larvae from each treatment. This is important to ensure the representativeness of the chemical data.
RESPONSE: All larvae remaining per tray at the end of the experimental period were sampled for chemical analyses. The reason for grouping larval samples per treatment at the end of the experimental period is that the low amount of sample does not allow for ensure enough material for a representative chemical analysis, as it was already stated in the former version. Explanation has been extended for clarification.
Line 92: The word "susbstituted" contains a typo. Correct to "substituted."
RESPONSE: Done
Lines 146–155: The statistical methods should be presented in a separate section for clarity and better organization.
RESPONSE: Statistical analysis is now presented in a separate subsection.
Discussion
Lines 205–315: The discussion is lengthy and covers multiple points without a clear structure. I suggest organizing the discussion into subtopics to improve clarity. Examples of subtopics could include: "Mortality and Growth," "Effects of Bread By-Product," "Impact of corn Silage and pH," and "Nutritional Composition of Larvae."
RESPONSE: We appreciate the reviewer´s recommendation. However, we consider that confusion should increase if subsections of Discussion were structured by mixing criteria (parameters affected and types of ingredients), as suggested. It is difficult to reorder Discussion in such terms since both “Mortality and growth” and “Larval composition” are treated jointly for all experimental treatments. Thus, considering that Discussion section is not extremely long (3 pages), we think it is better maintaining the actual structure.
Lines 237–243: The discussion mentions that the bread by-product did not alter growth performance and numerically increased survival, but it does not sufficiently emphasize its practical potential. I suggest adding a sentence about how the bread by-product can be an economical and efficient alternative to wheat grain, aligning with the sustainability objectives of the research.
RESPONSE: In the former version, the potential of bread by-product as ingredient in larval substrate was already mentioned in a summarising sentence, but it has been now completed mentioning the low cost and sustainability of this feed, as requested (now in L278)
Lines 245–265: Although the negative impact of Corn silage on performance is mentioned, I recommend adding a clear statement that the inclusion of Corn silage reduced substrate efficiency for larval growth. This would help avoid ambiguous interpretations regarding the viability of Corn silage as an alternative ingredient.
RESPONSE: The final sentence about the potential of maize silage has been rewritten (now in L350-351)
Line 248: Change "do not apparently match with differences observed in nutrient intake" to "do not appear to match the differences observed in nutrient intake."
RESPONSE: Done
Line 256: "necesssarily" contains a typo. Correct it to "necessarily."
RESPONSE: Done
Conclusions
The conclusion mentions that diets with wheat or bread by-product promoted similar growth, but it could emphasize that the bread by-product is an economically viable alternative, especially in comparison to wheat grain. I recommend adding a sentence highlighting the potential of the bread by-product as a sustainable and lower-cost option for feeding T. molitor.
RESPONSE: Actual writing indicates that bread by-product is a valid feed alternative because of its low price. A mention to the level of inclusion of bread by-product in substrate has been added.
Lines 335–338: Although Corn silage is mentioned as a viable alternative, the results showed that it had a negative impact on growth and feed efficiency. I suggest rephrasing the sentence to make it clear that Corn silage has limitations and that its use may only be viable if adjustments are made in formulation or if it is used at low levels.
RESPONSE: A mention to this has been added in conclusions.
Round 2
Reviewer 3 Report
Comments and Suggestions for Authors
The authors made the proposed modifications.